# Survey and molecular detection of Sri Lankan cassava mosaic virus in Thailand

Kingkan Saokham[1,2], Nuannapa Hemniam[3], Sukanya Roekwan[3], Sirikan Hunsawattanakul[4], Jutathip Thawinampan[3], Wanwisa Siriwan[3]*

1 Center of Agricultural Biotechnology, Kasetsart University, Nakhon Pathom, Thailand, 2 Center of Excellence on Agricultural Biotechnology: (AG-BIO/MHESI), Bangkok, Thailand, 3 Department of Plant Pathology, Faculty of Agriculture, Kasetsart University, Bangkok, Thailand, 4 Department of Agronomy, Faculty of Agriculture, Kasetsart University, Bangkok, Thailand

* wanwisa.si@ku.th

**Data Availability Statement:** All relevant data are within the paper and its Supporting information files.

## Abstract

Cassava plantations in an area of 458 hectares spanning five provinces along the Thailand–Cambodia border were surveyed from October 2018 to July 2019 to determine the prevalence of cassava mosaic disease (CMD) caused by Sri Lankan cassava mosaic virus (SLCMV) in the region. CMD prevalence was 40% in the whole area and 80% in Prachinburi, 43% in Sakaeo, 37% in Burium, 25% in Surin, and 19% in Sisaket provinces. Disease incidence of CMD was highest 43.08% in Sakaeo, followed by 26.78% in Prachinburi, 7% in Burium, 2.58% in Surin, and 1.25% in Sisaket provinces. Disease severity of CMD symptoms was mild chlorosis to moderate mosaic (2–3). The greatest disease severity was recorded in Prachinburi and Sakaeo provinces. Asymptomatic plants were identified in Surin (12%), Prachinburi (5%), Sakaeo (0.2%), and Buriram (0.1%) by PCR analysis. Cassava cultivars CMR-89 and Huai Bong 80 were susceptible to CMD. In 95% of cases, the infection was transmitted by whiteflies (*Bemisia tabaci*), which were abundant in Sakaeo, Buriram, and Prachinburi but were sparse in Surin; their densities were highest in May and June 2019. Nucleotide sequencing of the mitochondrial *cytochrome oxidase 1* (*mtCO1*) gene of whiteflies in Thailand revealed that it was similar to the *mtCO1* gene of Asia II 1 whitefly. Furthermore, the *AV1* gene of SLCMV—which encodes the capsid protein— showed 90% nucleotide identity with SLCMV. Phylogenetic analysis of completed nucleotide sequences of DNA-A and DNA-B components of the SLCMV genome determined by rolling circle amplification (RCA) indicated that they were similar to the nucleotide sequence of SLCMV isolates from Thailand, Vietnam, and Cambodia. These results provide important insights into the distribution, impact, and spread of CMD and SLCMV in Thailand.

## Background

Cassava is one of the most important food crops cultivated in Southeast Asia [1]. Approximately 55 million tons of cassava are produced in Southeast Asian countries per year, accounting for 30% of the global cassava production and valued at more than 10 million US dollars

**Funding:** The authors received funding by the Center of Excellence on Agricultural Biotechnology, Office of the Permanent Secretary, Ministry of Higher Education, Science, Research and Innovation. (AG-BIO/MHESI) and Thai Tapioca Development Institute (TTDI), Thailand. The funders had no role in study design, data collection and analysis, decision to publish, or preparation of the manuscript.

**Competing interests:** The authors have declared that no competing interests exist.

(USD). Thailand is one of the largest exporters of cassava products in the world and has a production capacity of approximately 31 million tons per year. In 2019, the export value of cassava from Thailand was 2.66 billion USD [2].

Cassava mosaic disease (CMD) caused by cassava mosaic geminivirus (CMV) is one of the most important diseases in Africa, as CMV is among the top 10 viruses affecting economically important crops [3]. The virus has a twinned icosahedral particle morphology and contains two genomic DNA components (DNA-A and DNA-B) [4]. The genome size of DNA-A and DNA-B components ranges from 2.7 to 3.0 kb [5]. CMV, which was first reported in Tanzania [6], belongs to the genus *Begomovirus* (family *Geminiviridae*) [7]. Plants affected by CMD have misshapen leaflets with foliar yellow or green mosaic patterns, curls, distortions, and mottling, which reduce the leaflet size and give a general appearance of stunting [8]. The virus is transmitted by whitefly (*Bemisia tabaci*) and via infected stem cuttings [9]; in Africa, these were shown to reduce the cassava yield by 35–60% and 55–77%, respectively [10]. Although nine CMV species have been reported across Africa and on islands in the Indian Ocean, only two occur in Asia, Indian cassava mosaic virus (ICMV) and Sri Lankan cassava mosaic virus (SLCMV) [11], with only the latter reported in Southeast Asia [12].

CMD emerged in Southeast Asia in 2015 [13]. In 2017, a survey of CMD was conducted in the Cambodian province of Stung Treng, which experienced an SCLMV outbreak despite its distant location from Ratanakiri [13]. Additionally, the Vietnam Academy of Agricultural Sciences Plant Protection Research Institute reported CMD in Tay Ninh province, where it damaged the established crop spanning more than 1200 ha of land in 2017 [14]. Based on a survey conducted in July and August 2018, the Department of Agriculture (DOA) of Thailand identified 22 plants with CMD symptoms in a 2.27-ha cassava plantation in Sisaket and Surin provinces in northeastern Thailand. According to the DOA, the infected plants were subsequently removed. However, CMD currently (2020–2021) affects more than 45,000 ha of the main cassava production area in Thailand [www.forecast-ppsf.doae.go.th] and has already been reported in South China and Laos [15, 16].

The current study presents the outcome of a CMD survey conducted across cassava plantations in five major provinces along the Thailand–Cambodia border from October 2018 to July 2019. We used appropriate and standardized procedures including PCR and DNA sequencing to detect CMD in the tested samples. The results of this survey provide an estimate of the spread and severity of CMD, data on the prevalence of the whitefly vector, and a classification of whitefly biotypes.

## Materials and methods

### Survey routes and sample collection

The survey was conducted from October 2018 to July 2019. Five major cassava-producing provinces of Thailand (Sisaket, Surin, Buriram, Sakaeo, and Prachinburi) located on the border with Cambodia were surveyed. An area of 458 ha planted with cassava (201 cassava fields) was used to sample cassava plantations in the five provinces (Fig 1). A total of thirty 3- to 6-month old cassava plants were randomly sampled from a 1-ha area of the plantation along two paths intersecting in an "X"; leaves were collected from the plants for PCR detection. The precise location of sampled plants was determined using the global positioning system (Compass Deluxe Navigation, a free application) [16]. The cultivar and age of sampled cassava plants, mode of CMD transmission, symptom severity, and number of whiteflies were noted following a previous study [17] with minor modifications. The size of whitefly populations was determined by counting the numbers of adult whiteflies on the five topmost leaves of each sampled plant [18].

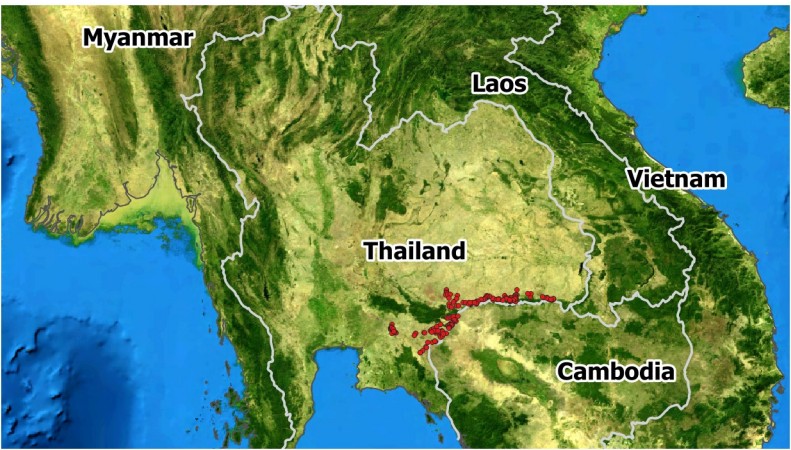

**Fig 1. Location of sampling sites of Sri Lankan cassava mosaic virus (SLCMV) in Thailand.** The survey was conducted from October 2018 to July 2019 on 201 cassava fields in Sisaket, Surin, Buriram, Sakaeo, and Prachinburi provinces.

## CMD prevalence, incidence, and symptom severity

Incidence of cassava mosaic disease was calculated as the disease prevalence and incidence using the following formulas:

Disease prevalence rate (%) was calculated using the following equation [19]:

$$Disease\ prevalence\ =\ \frac{Number\ of\ fields\ with\ visible\ symptoms}{Total\ number\ of\ fields\ observed} \times 100$$

Disease incidence (%) was calculated using the following equation:

$$Plant\ disease\ incidence\ (PDI)\ =\ \frac{(N-n)}{N} \times 100$$

where $N$ is the total number of observations, and $n$ is the total number of plants with no disease symptoms.

The severity of CMD symptoms was scored on a scale ranging from 1 to 5 (1 = no visible symptoms; 2 = mild chlorosis of the entire leaflet or mild distortion at the base of the leaflet, but overall green and healthy leaves; 3 = moderate mosaicism throughout the leaf, and narrowing and distortion of the lower one-third of the leaflet; 4 = severe mosaic and distortion of two-thirds of the leaflets, with general reduction in leaf size; and 5 = severe mosaicism, with distortion of the entire leaf) [20]. Disease severity index (DSI) was calculated using the following equation:

$$Plant\ disease\ severity\ (PDS) = \frac{[Sum\ (class\ frequency\ x\ score\ of\ rating\ class]}{Number\ of\ Plant\ Assessed} \times \frac{100}{Maximum\ Scale}$$

Two categories of infection were recognized and recorded as "cutting-borne" by the presence of symptoms on the lowest earliest-formed leaves) and "whitefly-borne" recognized by the presence of symptoms on upper leaves only (S2 Fig).

The latent infection rate (%) was calculated as follows:

$$Latent\ infection = \frac{Number\ of\ asymptomatic\ plants}{Total\ number\ of\ plants\ collected} \times 100$$

Adult whiteflies were collected from cassava fields located in Buriram, Sakaeo, and Surin provinces using an aspirator and transferred to 1.5-ml tubes containing 90% ethanol that were stored at −20˚C.

## Data analysis

A general linear model that considered location as fixed was used. Least square means for disease severity and number of whiteflies were estimated for each location and via cassava cultivars and were compared using Bonferroni t-tests. The data were analyzed using SAS software [21].

## DNA extraction and SLCMV detection

DNA was extracted from dried cassava leaves (20 mg) using the modified cetyl trimethylammonium bromide (CTAB) method [22]. Briefly, the leaves were crushed in CTAB buffer using metal beads and incubated at 65˚C for 30 min. The homogenized mixture was then added to 700 μl of chloroform:isoamyl alcohol (24:1), and DNA was precipitated using isopropanol alcohol for 3 h. The DNA pellet was washed twice with 70% ethanol and then dried at room temperature for approximately 30 min. The DNA was resuspended in water containing 100 μg/ml RNase (Thermo Fisher Scientific, Waltham, MA, USA) and stored at −20˚C. The quality and quantity of the isolated DNA were assessed by agarose gel electrophoresis and spectrophotometry [23].

To isolate DNA from whiteflies, five adults were randomly selected from among those collected at each location. Genomic DNA was isolated as previously described [24], with minor modifications. Briefly, each whitefly was crushed in lysis buffer (200 mM NaCl and 200 mM Tris-HCl [pH 8.0]) containing β-mercaptoethanol and proteinase K (10 mg/ml), and the mixture was incubated at 65˚C for 90 min. DNA was recovered by centrifugation.

To detect SLCMV, the *AV1* gene (encoding coat protein) was amplified from SLCMV-infected cassava leaf samples by PCR using sequence-specific primers (forward: 5′-GTT GAA GGT ACT TAT TCC C-3′ and reverse: 5′-TAT TAA TAC GGT TGT AAA CGC-3′) designed in our laboratory. A partial fragment of the mitochondrial cytochrome oxidase 1 (*mtCO1*) gene was amplified from whitefly DNA using the primers C1-J-2195 (5′-TTG ATT TTT TGG TCA TCC AGA AGT-3′) and L2-N-3014 (5′-TCC AAT GCA CTA ATC TGC CAT ATT A-3′) [25], yielding a 1258-bp product.

PCR amplification was performed in a 25-μl reaction volume containing 1× PCR buffer (PCR Biosystems, London, UK), 0.2 μM each of forward and reverse primers, and approximately 50 ng of the DNA template. The thermal cycling conditions for the *AV1* gene were as follows: initial denaturation at 94˚C for 5 min; 35 cycles of denaturation at 94˚C for 40 s, annealing at 55˚C for 40 s, and elongation at 72˚C for 40 s; and final elongation at 72˚C for 5 min. For the *mtCO1* gene, the thermal cycling conditions were as follows: initial denaturation at 94˚C for 5 min; 35 cycles of denaturation at 94˚C for 40 s, annealing at 52˚C (*mtCO1*) for 40 s, and elongation at 72˚C for 40 s; and final elongation at 72˚C for 5 min.

The amplified PCR products were separated on a 1% agarose gel alongside a 1-kb DNA ladder (Thermo Fisher Scientific) that was stained with RedSafe Nucleic Acid Staining Solution (iNtRON Biotechnology, Sangdaewon, South Korea) in 1× Tris–acetate–EDTA buffer. The

gels were visualized using a Gel Doc imaging system (Syngene, Frederick, MD, USA). Confirmed negative and positive controls were included in all assays.

## Complete genome characterization of the SLCMV isolate collected in Thailand

The Buri Ram province isolate was selected as a representative of CMV species diversity. The circular DNA of this isolate was obtained by rolling circle amplification (RCA) using phi29 DNA Polymerase (New England Biolabs, Ipswich, MA, USA), according to the manufacturer's instructions. The RCA product was digested with restriction endonucleases, and a ~2.7-kb fragment (the expected size of DNA-A and DNA-B fragments of SLCMV) was amplified from the digestion products. The DNA-A and DNA-B fragments were purified and cloned into the pGEM-T Easy vector (Promega, Madison, WI, USA) and then transformed into *Escherichia coli* strain DH5α cells by the heat shock method. The cloned inserts were sequenced in their entirety by primer walking (S1 Fig).

## Sequencing and phylogenetic analysis

Nucleotide sequences of the amplified fragments were searched in the National Center for Biotechnology Information database using BLAST (https://blast.ncbi.nlm.nih.gov/Blast.cgi). Multiple sequence alignment of the nucleotide sequences was performed using Molecular Evolutionary Genetics Analysis version X (MEGA X; http://www.megasoftware.net/) [26]. Phylogenetic trees were constructed in MEGA X using the neighbor-joining method with 1000 bootstrap replications.

## Results

### SLCMV diagnosis and symptom analysis

Field data collected during the survey are summarized in S1 Table. Survey locations were mapped using QGIS (Fig 1). The CMD prevalence data indicated that 80 cassava fields (40%) were infected with CMV. Severe CMD infection was identified in Prachinburi and Sakaeo provinces, whereas fields in Sisaket, Surin, and Buriram provinces showed mild infection. CMD prevalence was 80% in Prachinburi, 43% in Sakaeo, 37% in Burium, 25% in Surin, and 19% in Sisaket provinces (Table 1). Single leaves were collected from 6,120 cassava plants across 201 fields; 1434 were found to contain the virus by PCR analysis. The highest infection rate was in Sakaeo province (61.7%), followed by Prachinburi (20%), Surin (13%), Buriram (4.5%), and Sisaket (0.8%) provinces.

Infected cassava plants showed at least one of the typical CMD foliar symptoms such as green or yellow mosaic pattern, leaflet curling, and leaflet narrowing with distortion. Disease transmitted through infected stems caused symptoms in the whole plant, whereas transmission by whiteflies caused symptoms in only the top part of the plant (S2 Fig) [17]. Approximately 95% of the CMD incidence was attributable to whiteflies, with stem cuttings being responsible for 5% of infections. Stem cutting- and whitefly-borne infections were observed in the same plot. CMD symptoms typically appear 3–5 weeks after infection [27].

There was also a strong relationship between the mode of infection and whitefly populations. CMD was mainly spread by whiteflies, especially in Sakaeo, Prachinburi, and Buriram provinces, and there was no significant difference in disease incidence between Sisaket and Surin provinces (Table 1).

**Table 1. Field spread of CMD in Thailand based on a survey conducted across five provinces.**

| Province | Number of fields surveyed[a]/ Incidence[b] | Disease prevalence (%) [c] | Incidence (%) | Latent infection (%) [d] | Disease severity[e] | PCR positive (%) [f] | Source of infection (%) | | Average whitefly number/plant | Cultivar[g] |
|---|---|---|---|---|---|---|---|---|---|---|
| | | | | | | | Cutting | Whitefly | | |
| Prachinburi | 30 (96.2)/24 (67.4) | 80 | 26.78 | 5 | 2.54±0.19[a] | 20 | 11.6 | 88.4 | 3.37±1.51[a] | CMR-89, Rayong 72, Rayong 11, Rayong 9 and Huai Bong 80 |
| Sakaeo | 65 (148)/28 (59.2) | 43 | 43.08 | 0.2 | 2.15±0.13[a] | 61.7 | 1 | 99 | 5.19±1.02[a] | CMR-89, Rayong 72, Rayong 11, Kasetsart 50, Rayong 9 and Huai Bong 80 |
| Buriram | 30 (102.4)/11 (28.8) | 37 | 7 | 0.1 | 1.99±0.19[ab] | 4.5 | 0 | 100 | 4.09±1.51[a] | CMR-89, Rayong 72 and Kasetsart 50 |
| Surin | 44 (49)/11 (9.6) | 25 | 2.58 | 12 | 1.40±0.16[bc] | 13 | 56 | 44 | 0.32±1.24[b] | CMR-89, Rayong 72 and Kasetsart 50 |
| Sisaket | 32 (63)/6 (4.5) | 19 | 1.25 | 0 | 1.21±1.88[c] | 0.8 | 33.4 | 66.6 | 0.45±1.46[b] | CMR-89, Rayong 72 and Kasetsart 50 |

[a] Number of fields visited (number of area (ha)),

[b] Percentage number fields of incidence (number of area (ha)),

[c] Percentage of prevalence,

[d] Percentage of latent infection,

[e] Mean CMD severity scores are based on the standard 1 to 5 CMD scoring scale, where 1 = no symptoms and 5 = severe mosaic with distortion of entire leaf,

[f] Result of Positive PCR (Terry 1975) and

[g] Cultivars that showed infection are underlined.

## SLCMV symptom severity

Disease severity varied significantly between provinces (p<0.05). Disease severity was highest in Prachinburi and Sakaeo provinces and low in Sisaket province (Table 1). Plant age was correlated with the severity of disease symptoms. For example, 1- to 3-month-old infected plants had an average severity of 3.75 (moderate to severe mosaicism), whereas 5- to 7-month-old plants had an average severity score of 2.58 (mild chlorosis). Symptoms caused by cutting-borne disease were more severe than those caused by whitefly-borne infection.

Asymptomatic plants were detected in 22 (11%) cassava fields, especially in Surin (12%), Prachinburi (5%), and Sakaeo (0.2%) provinces. The lowest proportion of asymptomatic plants (0.1%) was in Buriram province.

We also surveyed cassava cultivars grown in the study area. Six cassava cultivars were identified on the surveyed route (Huai Bong 80, Rayong 9, Rayong 11, Rayong 72, Kasetsart 50, and CMR-89). Many farmers planted several cultivars in a single plot. The CMR-89 cultivar was the most common in the surveyed area, accounting for approximately 53% of the total area, followed by Rayong 72 (36%) and Kasetsart 50 (5%). Disease severity significantly differed (p<0.05) between cultivars: it was moderate in Huai Bong 80 and CMR-89 but low in Kasetsart 50 and Rayong 72 (Table 2). The main mode of transmission in all cassava cultivars was via whiteflies.

## Assessment of whitefly population size

Whitefly nymphs and adults were collected from the abaxial surface of the five topmost leaves of cassava plants. The nymphs had a flattened oval shape and resembled scaly insects. The average number of whiteflies significantly differed (p<0.05) between provinces. The

**Table 2. Proportion of cutting and whitefly-borne infection observed in cassava commercial cultivars in Thailand.**

| Cultivar | CMD incidence[a] (%) | Disease prevalence[b] (%) | Number of infected fields[d] /Number of fields surveyed[c] | Latent infection[e] (%) | CMD incidence of infection type[f] (%) | | Mean CMD severity[g] |
|---|---|---|---|---|---|---|---|
| | | | | | Cutting | Whitefly | |
| CMR-89 | 30.46 | 50 | 53/106 | 5.14 | 2.5 | 97.5 | 2.13±0.11[a] |
| Rayong 72 | 3.98 | 26.76 | 19/71 | 1.7 | 15 | 85 | 1.51±0.14[b] |
| Kasetsart 50 | 9.09 | 9.09 | 1/11 | 0 | 6.67 | 93.33 | 1.21±0.33[b] |
| Rayong 11 | 40 | 40 | 2/5 | 0 | 23.33 | 76.67 | 2.03±0.50[ab] |
| Rayong 9 | 1.67 | 25 | 1/4 | 0 | 50 | 50 | 1.50±0.55[ab] |
| Huaibong 80 | 35.56 | 100 | 4/4 | 0 | 15.63 | 84.37 | 3.24±0.56[a] |

[a] Percentage of disease incidence,

[b] Percentage of prevalence,

[c] Number of fields surveyed,

[d] Number of infected fields,

[e] Percentage of infection,

[f] Percentage of CMD incidence of infection type and

[g] Mean CMD severity scores are based on the standard 1 to 5 CMD scoring scale,

where 1 = no symptoms and 5 = severe mosaic with distortion of entire leaf,

[f] Result of Positive PCR [46].

population density was high in Sakaeo, Buriram, and Prachinburi provinces but low in Sisaket and Surin provinces (Table 1).

The survey was conducted from October 2018 to July 2019, which spans the cold season (October–February), summer season (March), and rainy season (July). The number of white-flies was high from May to July, with an average of 11.77, 10.6, and 6.59 per plant in May, June, and July, respectively; the average numbers were low from December to March.

## Amplification, sequencing, and phylogenetic analysis of the whitefly *mtCO1* gene

The nucleotide sequence of the *mtCO1* gene amplified from the DNA of whiteflies collected from Surin, Sakaeo, and Burirum provinces have been deposited in the DNA Data Bank of Japan (DDBJ) under accession numbers LC579572, LC579573, and LC579574, respectively. The *mtCO1* gene amplified from whiteflies in these provinces showed 99% sequence similarity to that of *B. tabaci* Asia II 1. Phylogenetic analysis of the *mtCO1* sequences of *B. tabaci* from Thailand showed that they grouped closely with reference sequences determined for a large number of Asia II 1 species collected from other regions in the world (Fig 2).

## PCR-based detection of CMV

PCR products were amplified from 1434 of 6120 samples collected from the five provinces using *AV1*-specific primers (S3 Fig). Of the 1434 PCR-positive samples, 61.7% were collected from Sakaeo, 20% from Prachinburi, 13% from Surin, 4.5% from Buriram, and 0.8% from Sisaket provinces. The PCR results also revealed that 205 samples harbored a latent infection; of these cases, samples from Surin and Buriram provinces showed the highest and lowest percent infection rates, respectively, whereas no latent infection was detected in plant samples from Sisaket province. Additionally, among the different cultivars, latent infection was detected in CMR-89 and Rayong 72 but not in Kasetsart 50, Rayong 9, Rayong 11, and Huaibong 80.

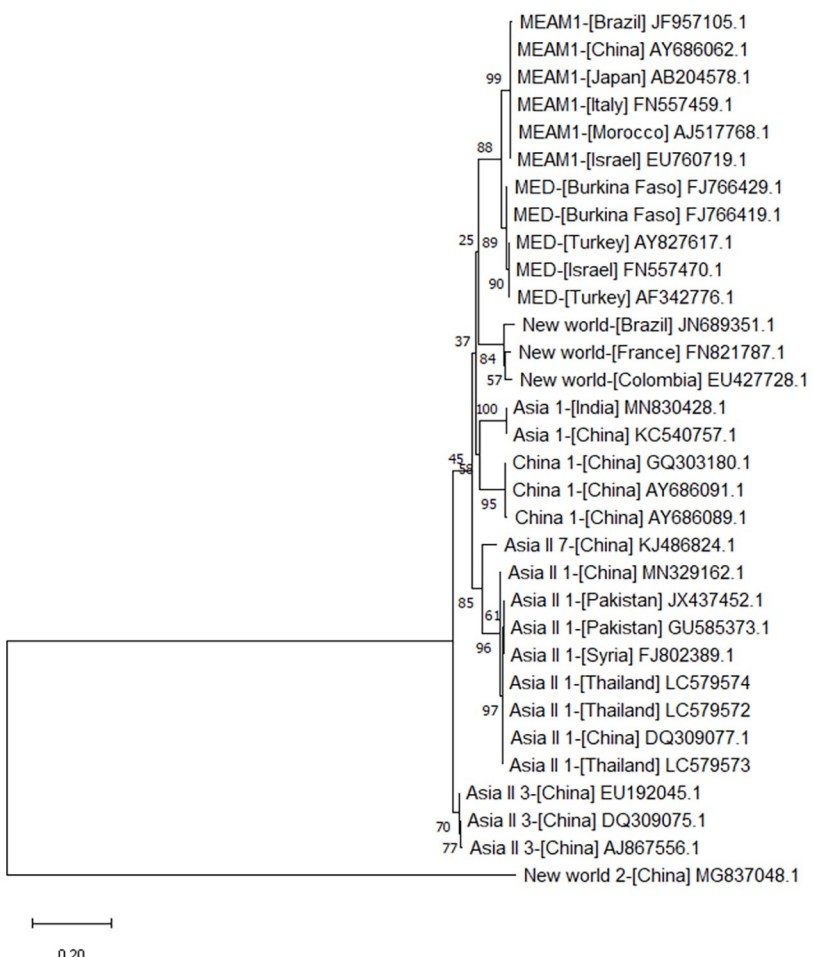

**Fig 2. Phylogenetic analysis of the nucleotide sequences of the *mtCOI* gene (length = 867 bp) of whitefly (*Bemisia tabaci*) collected from Surin (LC579572), Sakaeo (LC579573), and Buriram provinces (LC579574) in Thailand and that of other whitefly species.** Vertical distances are arbitrary. Horizontal distances are proportional to the calculated mutation distances. Numbers at nodes represent percent bootstrap confidence scores (1,000 replicates). The *mtCOI* gene of *B. tabaci* collected in this study clustered with that of Middle East Asia Minor 1 (MEAM1), *Mediterranean (MED)*, New World, Asia 1, China 1, Asia II 7, Asia II 1, Asia II 3, and New World 2 species available in GenBank.

## Whole-genome sequence of SLCMV

The complete genomic DNA sequence of the Burirum SLCMV isolate was obtained by RCA, and nucleotides sequences of DNA-A and DNA-B were submitted to the DDBJ under accession numbers LC586845 and LC588395, respectively. A BLAST search revealed that the DNA-A and DNA-B nucleotide sequences of the Burirum isolate were identical to those of previously characterized SLCMV isolates, with the highest sequence identity (99%) to isolates from Prachinburi (MN026159) [16].

We also conducted a phylogenetic analysis of the whole genome sequence of the Burirum SLCMV isolate. The phylogenetic tree indicated that the SLCMV isolates collected in our study belonged to the same species and were closely related to isolates from Vietnam (GenBank accession numbers LC312131 and LC312130); Cambodia (KT861468 and KT861469); Thailand (MN577578, MN954656, MT017511, MN026160, MN544647, MN577579, MN577575, MN577577, MN026159, MN577580, MN544648, and MN026161), China

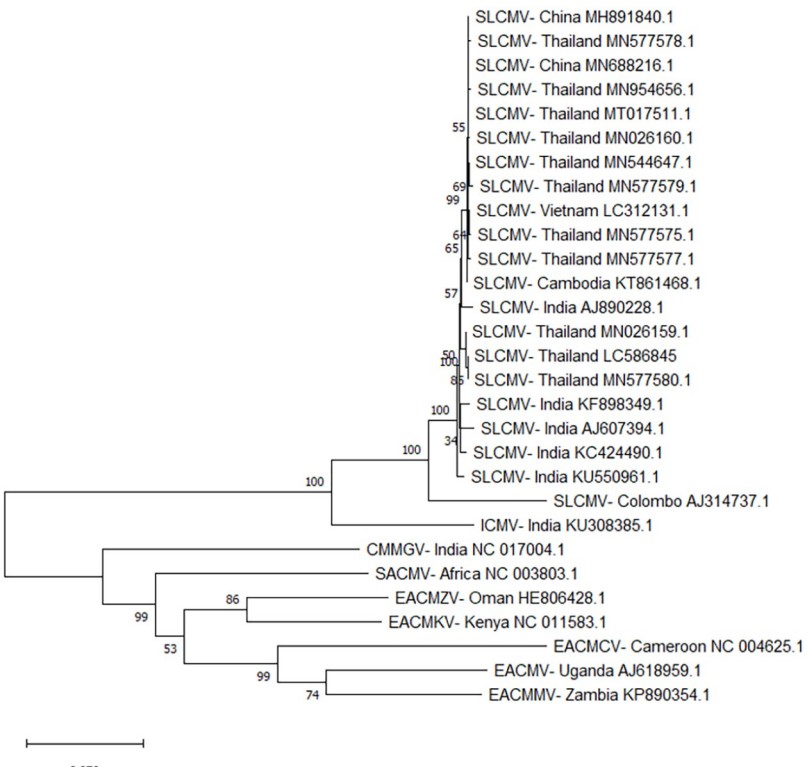

**Fig 3. Phylogenetic analysis of the nucleotide sequences of the DNA-A component of the Sri Lankan cassava mosaic virus (SLCMV) isolate collected from Buriram province (LC586845) and that of other CMV species.** Vertical distances are arbitrary. Horizontal distances are proportional to the calculated mutation distances. Numbers at nodes represent percent bootstrap confidence scores (1,000 replicates). ICMV, Indian cassava mosaic virus; CMMGV, Cassava mosaic Madagascar virus; SACMV, South African cassava mosaic virus; EACMZV, East African cassava mosaic Zanzibar virus; EACMKV, East African cassava mosaic Kenya virus; EACMCV, East African cassava mosaic Cameroon virus; EACMV, East African cassava mosaic virus; EACMMV, East African cassava mosaic Malawi virus.

(MH891840, MN688216, MN688251, and MN688217); and India (AJ890228, KF898349, AJ607394, KC424490, KU550961, MK404226, and AY730036) (Figs 3 and 4).

## Discussion

We surveyed CMD incidence and whitefly populations in an area where CMD has been previously reported as well as in new cassava plantations along the Thailand–Cambodia border. CMD was detected in some locations in five provinces where the disease was thought to have been eradicated by the DOA. Although the extent of the geographic area is an important factor affecting the eradication of CMD, other factors should also be considered such as planting distance, geographic location, mode of infection, and whitefly numbers [11].

A CMD outbreak was reported in Stung Treng province, Cambodia in 2016–2017 [13]. Stung Treng is located approximately 300 km from the Thailand border. Thailand is the main distributor of cassava to Cambodia, Laos, and other Southeast Asian countries [28]. CMD could rapidly spread through infected plant material transported across this region. We determined that the CMD outbreak in Thailand was initially caused by infected stem cuttings (primary infection source), and the second wave of the epidemic was caused by CMV transmitted via whiteflies. In Africa, CMD epidemics have been primarily driven by whiteflies [8, 11, 29, 30]; however, in Asia, whiteflies appear to play a secondary role in the spread of CMD.

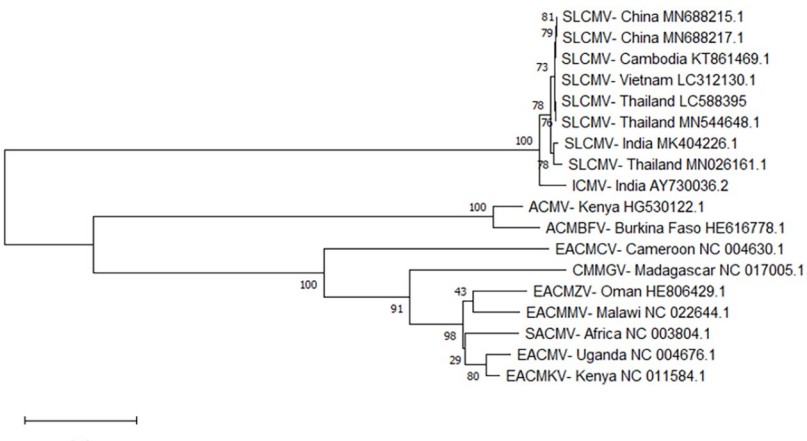

**Fig 4. Phylogenetic analysis of the nucleotide sequences of the DNA-B component of the SLCMV isolate collected from Buriram province (LC588395) and that of other CMV species.** Vertical distances are arbitrary. Horizontal distances are proportional to the calculated mutation distances. Numbers at nodes represent percent bootstrap confidence scores (1,000 replicates). ICMV, Indian cassava mosaic virus; ACMV, African cassava mosaic virus; ACMBFV, African cassava mosaic Burkina Faso virus; EACMCV, East African cassava mosaic Cameroon virus; CMMGV, Cassava mosaic Madagascar virus; EACMZV, East African cassava mosaic Zanzibar virus; EACMMV, East African cassava mosaic Malawi virus; SACMV, South African cassava mosaic virus; EACMV, East African cassava mosaic virus; EACMKV, East African cassava mosaic Kenya virus.

Nonetheless, the results of epidemiologic studies in Ivory Coast, Kenya, and Uganda support our speculation that the spread of CMD into and within the experimental cassava cultivation area was directly related to the number of adult whiteflies present and to the incidence of CMD, as determined by the locality or administrative district where the trials were carried out and where CMD dissemination was widespread. Any subsequent spread of CMD is attributable to the viruliferous whiteflies moving between or within planting areas after acquiring the virus from cassava plants grown from infected cuttings or infected by whiteflies during growth [29, 31].

The region bordering Thailand and Cambodia is rich in forests and high mountains, which act as a natural barrier to the movement of whiteflies. It is possible that the incidence of CMD in Thailand near the Cambodian border was caused by the exchange of infected cassava planting material among local populations.

Disease incidence is related to fluctuations in the whitefly population due to environmental factors such as rainfall, wind, and temperature [10]. We found a large number of whiteflies in Prachinburi and Sakaeo provinces, consistent with the disease incidence rates in these provinces (Table 1). Furthermore, whitefly population size impacts the spread of CMD, as whiteflies can travel distances of up to 100 km a year [8], with an estimated flight speed of approximately 0.2 m/s [10]. During its life cycle of approximately 30–40 days, a female whitefly lays up to 300 eggs on the abaxial surface of leaves [32]. Temperature, humidity, and rainfall influence the population size of adult whiteflies. Conditions that are conducive to an increase in whitefly numbers include temperatures <35°C and a relative humidity of approximately 65–73% [15]. Whitefly density was the highest in May, followed by June and July. May marks the beginning of the rainy season in Thailand, with temperatures <30°C and approximately 64% relative humidity [33]. We therefore propose that farmers should be persuaded to modify their traditional planting practices—which include planting soon after the onset of the rainy season—to avoid high disease incidence caused by abundant whitefly populations.

We found that most of the infections were caused by whitefly, which influenced the spread of CMD in the surveyed area. CMV transmitted by whiteflies has caused CMD not only in Thailand but also in Africa since the 16th century [34]. Understanding the ecological and biological characteristics of whiteflies can aid the prediction of future CMD epidemics according to weather data, thereby facilitating disease management [17].

Disease severity is affected by virus strain, plant age, plant genotype, and environmental conditions [35]. In CMD-resistant varieties, the appearance of symptoms in leaves is influenced to a greater extent by cooler temperatures than by hot weather [36]. Moreover, symptoms are exacerbated in plants regenerated from infected planting material. In this study, CMR-89 was susceptible to CMD (>70% disease incidence) and showed the highest disease severity among the seven tested cultivars. Although CMR-89 is not a DOA-certified cultivar, it is grown in approximately 22% of cassava plantations in Thailand (Office Agricultural Economics 2018). In one study that screened CMD resistance in cassava cultivars by grafting, CMR-89 and Rayong 11 were found to be susceptible to CMD, whereas Kasetsart 50, Rayong 72, and Huai Bong 60 showed moderate resistance [37]. Discontinuing the cultivation of CMR-89 and promoting the cultivation of CMD-tolerant or moderately resistant cultivars is critical for controlling this disease.

The pattern of CMD spread differed depending on the mode of transmission. Most cassava plants infected by whiteflies were located at the edge of the plot, with the infection then spreading inward. Whitefly density was especially high in newly planted cassava stands located close to mature cassava plants. Similar cases have been reported in several countries in East and West Africa, where new cassava plantings were colonized by whitefly populations immigrating from older cassava stands in the area. The immigrant whiteflies reproduced and reached their peak population size within a few months, and before the population declined adults dispersed to younger cassava plants [38–40]. Thus, the whitefly count can be useful for predicting and controlling the spread of CMD, and farmers should frequently monitor their cassava plants and whitefly populations.

CMD has been reported in several Southeast Asian countries. SLCMV has been detected in cassava fields in Thailand, and similar viruses have been reported in Cambodia, Vietnam, and China [13, 16, 41]. CMD was reported in Thailand in 2018 after its occurrence in Cambodia and Vietnam. The viral strain identified in this study has the same origin as that first reported in Ratanakiri, Cambodia, and other studies conducted in Thailand [16, 42].

The Rep protein encoded by DNA-A of the Burirum SLCMV isolate had seven additional amino residues at its C-terminal end. This 7-amino-acid motif is essential for the accumulation of the Rep protein and virulence of SLCMV [12]. The genomes of SLCMV isolates from Southern India, Sri Lanka, and Southeast Asia were not recombinant but harbored a point mutation [16]. Additionally, SLCMV isolates from Southeast Asia, China, and India clustered together in a separate group from the original SLCMV isolate from Colombo, Sri Lanka (AJ314737) [4] (Fig 3). Further investigation is needed to determine the host range of SLCMV, clarify the mechanisms of transreplication of its DNA components, and identify the genetic determinants of symptoms.

According to the phylogenetic analysis, the partial coding sequence of *mtCO1* of *B. tabaci* from Thailand was classified as an Asia II 1 cryptic species. Asia II 1 whiteflies readily transmit SLCMV, whereas Middle East Asia Minor 1 and Mediterranean whiteflies are poor vectors of the virus [43]. Thus, the potential for virus transmission is associated with the virus and whitefly species. In cassava fields in southern Vietnam, multiple indigenous whitefly species have been identified including Asia 1, Asia II 1, and Asia II 6 [44]. An Asia II 1 cryptic species was shown to efficiently transmit cotton leaf curl Multan virus (CLCuMuV) [45]. Therefore, in an area where Asia II 1 species are predominant, the implementation of phytosanitary measures

and rouging may not be sufficient for limiting the spread of the virus. Further research is needed on the control virus transmission by indigenous whitefly species to facilitate the development of durable control strategies. The Asia II 1 species shows a very high level of insecticide resistance [46], which must be taken into account in whitefly population management.

Based on our results, we propose the following basic approaches for controlling the outbreak and spread of CMD: 1) educate farmers and agricultural extension officers about CMD, including how to distinguish CMD symptoms from mineral deficiency or herbicide toxicity; 2) develop CMD-resistant cassava varieties and cultivate them on a sufficiently large scale; 3) practice phytosanitary techniques such as the use of CMD-free planting material and removal (rouging) of diseased plants; and 4) avoid planting cassava varieties susceptible to CMD such as CMR-89 and Rayong 11, especially in high-risk areas.

## Conclusion

We surveyed the spread of CMD in five major cassava-producing provinces of Thailand along the border with Cambodia. This is the first survey to report patterns of CMD spread, disease incidence and severity, and whitefly density in Thailand. This information will aid the development of disease management strategies to reduce the spread of CMD in affected areas. Although conducting surveys is costly and time-consuming, the information that is obtained is critical for disease epidemiology.

## Supporting information

**S1 Fig. Schematic showing the whole-genome sequence of Sri Lankan cassava mosaic virus (SLCMV) using the primer walking approach.**
(PDF)

**S2 Fig. Symptoms of cassava mosaic disease in cassava-infected plants.** A) CMD transmitted through infected stem, B) CMD transmitted by *B. tabaci.*
(TIF)

**S3 Fig. PCR products of the *AV1* gene using SLCMV specific primers.** DNA gel electrophoresis of PCR amplification from cassava samples.
(TIF)

**S1 Table. Field data collected during the survey, including field location, cassava cultivar, mode of infection, and disease severity.**
(XLSX)

**S2 Table. Dataset of disease severity in cassava cultivars during the survey.**
(XLSX)

## Author Contributions

**Conceptualization:** Nuannapa Hemniam, Wanwisa Siriwan.

**Data curation:** Kingkan Saokham, Nuannapa Hemniam, Sukanya Roekwan, Sirikan Hunsawattanakul, Jutathip Thawinampan, Wanwisa Siriwan.

**Formal analysis:** Kingkan Saokham, Nuannapa Hemniam, Sirikan Hunsawattanakul, Jutathip Thawinampan, Wanwisa Siriwan.

**Funding acquisition:** Wanwisa Siriwan.

**Investigation:** Wanwisa Siriwan.

**Methodology:** Kingkan Saokham, Nuannapa Hemniam, Sukanya Roekwan, Sirikan Hunsa-wattanakul, Jutathip Thawinampan.

**Project administration:** Kingkan Saokham.

**Resources:** Kingkan Saokham.

**Software:** Kingkan Saokham, Nuannapa Hemniam.

**Supervision:** Wanwisa Siriwan.

**Validation:** Kingkan Saokham, Wanwisa Siriwan.

**Visualization:** Wanwisa Siriwan.

**Writing – original draft:** Kingkan Saokham, Wanwisa Siriwan.

**Writing – review & editing:** Wanwisa Siriwan.

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
