## [Decision Letter · Decision Letter 0]

2 Jul 2021

PONE-D-21-16747

Survey and Molecular Detection of Sri Lankan Cassava Mosaic Virus in Thailand

PLOS ONE

Dear Dr. Siriwan,

Thank you for submitting your manuscript to PLOS ONE. After careful consideration, we feel that it has merit but does not fully meet PLOS ONE’s publication criteria as it currently stands. Therefore, we invite you to submit a revised version of the manuscript that addresses the points raised during the review process.

We look forward to receiving your revised manuscript.

Kind regards,

Hanu R Pappu

Academic Editor

PLOS ONE

Journal Requirements:

3. In your Methods section, please provide additional location information, including geographic coordinates for the data set if available.

"This study was supported by the Center of Excellence on Agricultural Biotechnology,

Science and Technology Postgraduate Education and Research Development Office, Office

of Higher Education Commission, Ministry of Education (AG-BIO/PERDO-CHE) and Thai

Tapioca Development Institute (TTDI), Thailand."

"The authors received no specific funding for this work. The funders had no role in study

design, data collection and analysis, decision to publish, or preparation of the

manuscript."

6. Please include your tables as part of your main manuscript and remove the individual files. Please note that supplementary tables (should remain/ be uploaded) as separate "supporting information" files.

Reviewers' comments:

Reviewer's Responses to Questions

**Comments to the Author**

1. Is the manuscript technically sound, and do the data support the conclusions?

Reviewer #1: Yes

Reviewer #2: Partly

Reviewer #3: Partly

2. Has the statistical analysis been performed appropriately and rigorously? 

Reviewer #1: Yes

Reviewer #2: N/A

Reviewer #3: No

3. Have the authors made all data underlying the findings in their manuscript fully available?

Reviewer #1: No

Reviewer #2: Yes

Reviewer #3: Yes

4. Is the manuscript presented in an intelligible fashion and written in standard English?

Reviewer #1: Yes

Reviewer #2: No

Reviewer #3: Yes

5. Review Comments to the Author

Reviewer #1: Abstract:

“Cassava plantations in an area of 458 ha spanning five province….” do not use unit abbreviations in abstract.

“Disease severity was generally scored as 2–3….” Based on which scale? Mention either disease severity index value or disease incidence.

“Prachinburi but was sparse in Surin, with the largest populations observed in May and June…” Mention year with months.

Materials and Methods section describes more about disease incidence and its prevalence. The author has used formulas from incidence and prevalence to assess the disease score. However, the “SLCMV prevalence and symptom severity” heading under result section is focusing on the data based on disease severity index (DSI) and prevalence. It is better to add DSI formula under the materials and methods. Further use separate heading for DSI data under result section.

Author should provide supplementary data including gel images for PCR based detection of CMV.

Page 11. “At the first inspection, we found four plants infected by stem cuttings; in the second visit, 10 plants show CMD symptoms on the top 4–6 leaves; and in the last inspection, 15 plants were infected by whiteflies. This phenomenon was observed not only in Surin but also in other

provinces included in the survey….” Provide supplementary data including images or pictures collected for this CMD infection on leaves.

Overall, there are minor grammatical problems throughout the manuscript. Read the manuscript thoroughly before re-submission.

Reviewer #2: Abstract:

Sentence 7 should read: In 95% of cases, the infection was due to virus transmission by whitefly……..

Sentence 9: “Furthermore, the AV1 gene ………….” Please clarify the AV1 gene of which virus?

Background:

Paragraph 1:

The first sentence needs to be referenced.

It’s not clear from the last sentence in the paragraph whether the export value of 2.66 billion USD refers to Thailand.

Paragraph 2:

The first sentence is not clearly expressed. It can be changed to read “ Cassava mosaic disease (CMD) caused by cassava mosaic geminivirus (CMV) is one of the most important diseases in Africa with CMV being among the top 10 viruses affecting economically important crops.”

The second sentence must be followed by the fourth sentence. The authors must then say something about the genome size of CMV.

In sentence 5, authors must state the mode of transmission of the virus by whitefly.

Last sentence: Replace word “including” with a semi-colon. Indian cassava mosaic virus must be followed by virus acronym in brackets.

Paragraph 3:

The first sentence does not make sense. CMD emerged 2016-2017 after being reported in North East Cambodia in 2015! This disparity needs to be reconciled

Materials and Methods:

A map to show the areas survey will be good.

DNA extraction and SLCMV detection:

It is not clear whether the PCR thermal cycling conditions described are for detecting the SLCMV AV1 gene or the whitefly mtCO1 gene. Authors must differentiate between the conditions for the two genes.

Complete genome characterization of the SLCMV isolate …..

The authors must explain why the Burian province isolate was selected as the representative sample for whole genome sequencing.

Results:

SLCMV diagnosis and symptom analysis:

First sentence: The Supplementary table S1 is showing primer sequences not survey data as stated in the first sentence. This needs to be corrected.

Second paragraph:

Authors can show pictures of the difference between cutting-borne and whitefly-borne symptoms. A picture speaks a thousand words.

Third paragraph:

The whole paragraph needs to be re-written. It is not clearly explained how the authors determined that the first symptoms were due to cutting-borne infections and the later ones were due to white-borne infections.

Assessment of whitefly population size:

Paragraph 2:

Authors need to explain what they mean by “two month old plants facilitated whitefly counting”

Discussion:

Paragraph 1:

First sentence should read: “We surveyed for CMD incidence and whitefly population in an area where CMD had previously been reported…….”

Paragraph 2:

Second sentence should read: “Steng Teng is situated approximately ………”

Paragraph 6:

First sentence is stating the very obvious. What is the point the authors are making here?

Reviewer #3: Specific comments

Page 16: “SLCMV prevalence and symptom severity”. Authors have carried out PCR diagnoses for the presence of viruses in their field-collected samples. They have used pair of primer specific for SLCMV. However, PCR with SLCMV specific primer could also amplify other closely related begomoviruses. Wondering, how do they rule out this?

Page 19: “Whole-genome sequence of SLCMV”. Out of 6120 samples collected from different provinces, they have detected the presence of SLCMV in 1434 samples by PCR using AV1-specific primers. Later, the authors have mentioned that they have cloned and sequenced full-length viral genome. How many full-length viruses did they cloned and sequenced? How many samples did they choose for each province for full-length virus cloning and sequencing?

6. PLOS authors have the option to publish the peer review history of their article (what does this mean?). If published, this will include your full peer review and any attached files.

Reviewer #1: No

Reviewer #2: No

Reviewer #3: No

---

## [Author Response · Author response to Decision Letter 0]

3 Aug 2021

Journal Requirements:

No permit was required for cassava fields because cassava is economic plant in Thailand. In case, researches involve in National Park, wild animal, special conservation plant species and Plant varieties protection etc. 

Cassava mosaic disease was emerging disease in Thailand during our survey. Department of Agriculture (DOA) needed partner to survey and identified disease, we were one group of researches to join them. However, we needed to report DOA every time after disease was found. The survey data also recorded by DOA for reporting FAO. 

3. In your Methods section, please provide additional location information, including geographic coordinates for the data set if available.

 We provided additional location information including latitude and longitude. This information presented as Supplementary data Table. 

"This study was supported by the Center of Excellence on Agricultural Biotechnology,

Science and Technology Postgraduate Education and Research Development Office, Office

of Higher Education Commission, Ministry of Education (AG-BIO/PERDO-CHE) and Thai

Tapioca Development Institute (TTDI), Thailand."

"The authors received no specific funding for this work. The funders had no role in study

design, data collection and analysis, decision to publish, or preparation of the

manuscript."

The Acknowledgments Section already remove from revised Manuscript. 

Please add Funding Statement section of the online submission form as 

“The authors received funding by the Center of Excellence on Agricultural Biotechnology,

Science and Technology Postgraduate Education and Research Development Office, Office of Higher Education Commission, Ministry of Education (AG-BIO/PERDO-CHE) and Thai Tapioca Development Institute (TTDI), Thailand. The funders had no role in study design, data collection and analysis, decision to publish, or preparation of the

manuscript."

6. Please include your tables as part of your main manuscript and remove the individual files. Please note that supplementary tables (should remain/ be uploaded) as separate "supporting information" files.

We already included tables in the Revised Manuscript and removed the individual files as recommend. 

Reviewers' comments:

Reviewer's Responses to Questions

Comments to the Author

1. Is the manuscript technically sound, and do the data support the conclusions?

Reviewer #1: Yes

Reviewer #2: Partly

Reviewer #3: Partly

2. Has the statistical analysis been performed appropriately and rigorously?

Reviewer #1: Yes

Reviewer #2: N/A

Reviewer #3: No

3. Have the authors made all data underlying the findings in their manuscript fully available?

Reviewer #1: No

Reviewer #2: Yes

Reviewer #3: Yes

4. Is the manuscript presented in an intelligible fashion and written in standard English?

Reviewer #1: Yes

Reviewer #2: No

Reviewer #3: Yes

5. Review Comments to the Author

Reviewer #1: Abstract:

“Cassava plantations in an area of 458 ha spanning five province….” do not use unit abbreviations in abstract.

The recommend has fix as mention. 

“Disease severity was generally scored as 2–3….” Based on which scale? 

The severity score based on 1-5 scale 

1=no visible symptoms; 

2=mild chlorosis of the entire leaflet or mild distortion at the base of the leaflet, but overall green and healthy leaves; 

3=moderate mosaicism throughout the leaf, and narrowing and distortion of the lower one-third of the leaflet; 

4=severe mosaic and distortion of two-thirds of the leaflets, with general reduction in leaf size; 

5=severe mosaicism, with distortion of the entire leaf

Mention either disease severity index value or disease incidence.

Disease incidence of each province was added in the abstract as recommend. 

“Prachinburi but was sparse in Surin, with the largest populations observed in May and June…” Mention year with months.

The recommend has fix as mention. 

Materials and Methods section describes more about disease incidence and its prevalence. The author has used formulas from incidence and prevalence to assess the disease score. However, the “SLCMV prevalence and symptom severity” heading under result section is focusing on the data based on disease severity index (DSI) and prevalence. It is better to add DSI formula under the materials and methods. Further use separate heading for DSI data under result section.

Done as recommended from reviewer

Author should provide supplementary data including gel images for PCR based detection of CMV.

The representative of gel images for PCR based detection of CMD was provided in supplementary data. 

Page 11. “At the first inspection, we found four plants infected by stem cuttings; in the second visit, 10 plants show CMD symptoms on the top 4–6 leaves; and in the last inspection, 15 plants were infected by whiteflies. This phenomenon was observed not only in Surin but also in other provinces included in the survey….” Provide supplementary data including images or pictures collected for this CMD infection on leaves.

We had technical problem for images and pictures collection of this incidence. We determined to remove this paragraph to avoid suspicion issues that may happen. 

Overall, there are minor grammatical problems throughout the manuscript. Read the manuscript thoroughly before re-submission.

Reviewer #2: Abstract:

Sentence 7 should read: In 95% of cases, the infection was due to virus transmission by whitefly……..

The recommend has fix as mention. 

Sentence 9: “Furthermore, the AV1 gene ………….” Please clarify the AV1 gene of which virus?

The recommend has fix as mention. There was identified AV1 gene of SLCMV.

Background:

Paragraph 1:

The first sentence needs to be referenced.

The reference was added 

It’s not clear from the last sentence in the paragraph whether the export value of 2.66 billion USD refers to Thailand. 

Yes, I was referred to Thailand. Sentence had revised to be clear.

Paragraph 2:

The first sentence is not clearly expressed. It can be changed to read “Cassava mosaic disease (CMD) caused by cassava mosaic geminivirus (CMV) is one of the most important diseases in Africa with CMV being among the top 10 viruses affecting economically important crops.”

The recommend has fix as mention. 

The second sentence must be followed by the fourth sentence. The authors must then say something about the genome size of CMV.

The recommend has fix as mention. 

In sentence 5, authors must state the mode of transmission of the virus by whitefly.

Last sentence: Replace word “including” with a semi-colon. Indian cassava mosaic virus must be followed by virus acronym in brackets.

The recommend has fix as mention. 

Paragraph 3:

The first sentence does not make sense. CMD emerged 2016-2017 after being reported in North East Cambodia in 2015! This disparity needs to be reconciled

The recommend has fix as mention. 

Materials and Methods:

A map to show the areas survey will be good.

Surveyed map was added in the manuscript.

DNA extraction and SLCMV detection:

It is not clear whether the PCR thermal cycling conditions described are for detecting the SLCMV AV1 gene or the whitefly mtCO1 gene. Authors must differentiate between the conditions for the two genes.

The PCR thermal cycling conditions of SLCMV detection and mtCO1 gene were separately describe as recommended. 

Complete genome characterization of the SLCMV isolate …..

The authors must explain why the Burian province isolate was selected as the representative sample for whole genome sequencing.

The main reasons why we identified whole genome of SLCMV in Buri Ram province. First, during that time SLCMV Buri Ram province not yet identified whole genome, while whole genome of SLCMV from Sisaket, Surin, Sakaeo and Prachin Buri provinces were identified by another groups. Second, SLCMV infected plants were discover by our survey team before announcement from Department of Agriculture (DOA), Thailand. 

Results:

SLCMV diagnosis and symptom analysis:

First sentence: The Supplementary table S1 is showing primer sequences not survey data as stated in the first sentence. This needs to be corrected.

The recommend has fix as mention. 

Second paragraph:

Authors can show pictures of the difference between cutting-borne and whitefly-borne symptoms. A picture speaks a thousand words.

A picture of the difference between cutting-borne and whitefly-borne symptoms was provided as Fig.??

Third paragraph:

The whole paragraph needs to be re-written. It is not clearly explained how the authors determined that the first symptoms were due to cutting-borne infections and the later ones were due to white-borne infections.

We made a decision to remove this paragraph to avoid misunderstand from readers. 

Assessment of whitefly population size:

Paragraph 2:

Authors need to explain what they mean by “two month old plants facilitated whitefly counting”

In this survey, we counted B. tabaci in cassava fields that were 2-4 months old. According to previous researches were presented below.

B. tabaci populations have consistently been reported to peak three to six months after planting, before declining more or less rapidly to a relatively low level for the remainder of the crop life [1]. Whitefly population also reduce when plant more matured. They recommenced to counting whitefly population in cassava plant at 30-60 days after planting. One of another research presented that the association between number of cassava leaf and B. tabaci population change, with increases in number of whiteflies occurring during periods of rapid cassava growth [1]. 

Discussion:

Paragraph 1:

First sentence should read: “We surveyed for CMD incidence and whitefly population in an area where CMD had previously been reported…….”

The recommend has fix as mention. 

Paragraph 2:

Second sentence should read: “Steng Teng is situated approximately ………”

The recommend has fix as mention. 

Paragraph 6:

First sentence is stating the very obvious. What is the point the authors are making here?

We removed that sentence to avoid miss understanding to readers. The recommend has fix as mention. 

Reviewer #3: Specific comments

Page 16: “SLCMV prevalence and symptom severity”. Authors have carried out PCR diagnoses for the presence of viruses in their field-collected samples. They have used pair of primer specific for SLCMV. However, PCR with SLCMV specific primer could also amplify other closely related begomoviruses. Wondering, how do they rule out this?

We also wondering if they had difference CMV stains outbreak in Thailand during this survey, SLCMV specific primers could not identified. We make sure by analyzed partial DNA sequencing. The samples for analyzed DNA sequencing random sampling from 10% of samples in each province. The result of DNA sequencing show all were SLCMV. In addition, we are continue monitoring CMD situation in Thailand, Cambodia, Vietnam and China. Lucky, only SLCMV has been report. 

Page 19: “Whole-genome sequence of SLCMV”. Out of 6120 samples collected from different provinces, they have detected the presence of SLCMV in 1434 samples by PCR using AV1-specific primers. Later, the authors have mentioned that they have cloned and sequenced full-length viral genome. How many full-length viruses did they cloned and sequenced? How many samples did they choose for each province for full-length virus cloning and sequencing?

We cloned one full length virus genome and sequenced. 

6. PLOS authors have the option to publish the peer review history of their article (what does this mean?). If published, this will include your full peer review and any attached files.

Do you want your identity to be public for this peer review? For information about this choice, including consent withdrawal, please see our Privacy Policy.

Reviewer #1: No

Reviewer #2: No

Reviewer #3: No

Reference 

1. Legg J. Bemisia Tabaci: The Whitefly Vector of Cassava Mosaic Geminiviruses in Africa: An Ecological Perspective. African Crop Science Journal (ISSN: 1021-9730) Vol 2 Num 4. 1994;2.

---

## [Editor Report · Decision Letter 1]

28 Sep 2021

Survey and Molecular Detection of Sri Lankan Cassava Mosaic Virus in Thailand

PONE-D-21-16747R1

Dear Dr. Siriwan,

We’re pleased to inform you that your manuscript has been judged scientifically suitable for publication and will be formally accepted for publication once it meets all outstanding technical requirements.

Kind regards,

Hanu R Pappu

Academic Editor

PLOS ONE
---

## [Editor Report · Acceptance letter]

1 Oct 2021

PONE-D-21-16747R1 

Survey and Molecular Detection of Sri Lankan Cassava Mosaic Virus in Thailand 

Dear Dr. Siriwan:

I'm pleased to inform you that your manuscript has been deemed suitable for publication in PLOS ONE. Congratulations! Your manuscript is now with our production department. 

Kind regards, 

on behalf of

Dr. Hanu R Pappu 

Academic Editor

PLOS ONE